# Beyond In-Place Corruption: Insertion and Deletion in Denoising Probabilistic Models

**Daniel D. Johnson** [1]   **Jacob Austin** [1]   **Rianne van den Berg** [1]   **Daniel Tarlow** [1]

## Abstract

Denoising diffusion probabilistic models (DDPMs) have shown impressive results on sequence generation by iteratively corrupting each example and then learning to map corrupted versions back to the original. However, previous work has largely focused on *in-place corruption*, adding noise to each pixel or token individually while keeping their locations the same. In this work, we consider a broader class of corruption processes and denoising models over sequence data that can insert and delete elements, while still being efficient to train and sample from. We demonstrate that these models outperform standard in-place models on an arithmetic sequence task, and that when trained on the text8 dataset they can be used to fix spelling errors without any fine-tuning.

## 1. Introduction

Although autoregressive models are generally considered state of the art for language modeling, machine translation, and other sequence-generation tasks (Raffel et al., 2020; van den Oord et al., 2016), they must process tokens one at a time, which can make generation slow. As such, significant research effort has been put into non-autoregressive models that allow for parallel generation (Wang & Cho, 2019; Ghazvininejad et al., 2019). Recently, denoising diffusion probabilistic models (DDPMs) (Sohl-Dickstein et al., 2015) have shown impressive results in a variety of domains (Chen et al., 2020; Ho et al., 2020; Hoogeboom et al., 2021; Austin et al., 2021), in some cases achieving comparable results to autoregressive models with far fewer steps. In these models, a *forward process* iteratively corrupts the data towards a noise distribution, and a generative model is trained to learn the reverse denoising process. However, these models

---

[1]Google Research, Brain team. Correspondence to: Daniel D. Johnson <ddjohnson@google.com>.

Third workshop on *Invertible Neural Networks, Normalizing Flows, and Explicit Likelihood Models* (ICML 2021). Copyright 2021 by the author(s).

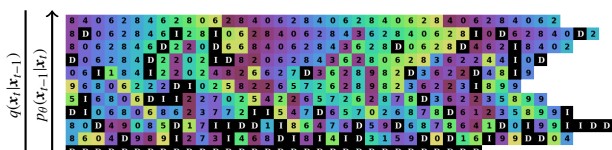

*Figure 1.* Generating an arithmetic sequence by denoising with insertion and deletion over ten steps, showing $x \mod 100$ with color and $x \mod 10$ with text. 'D' denotes deletion and 'I' insertion according to the fixed forward process $q(\boldsymbol{x}_t|\boldsymbol{x}_{t-1})$. This sequence was generated by the learned reverse process $p_\theta(\boldsymbol{x}_{t-1}|\boldsymbol{x}_t)$.

share one limitation: the corruption process always modifies sequence elements in-place. While convenient, this choice introduces strong constraints that limit the efficacy of the generative denoising process. For example, if the model makes a mistake and places a word or phrase in the wrong place, it cannot easily compensate.

For sequence-to-sequence tasks, the Levenshtein transformer (Gu et al., 2019) and Insertion-Deletion transformer (Ruis et al., 2020) address this limitation by performing insertion and deletion operations. However, these models were not designed as purely generative models, and do not in general allow estimation of sample log-likelihoods through both the insertion and deletion phases.

In this work, we integrate insertion and deletion into the DDPM framework, generalizing multinomial diffusion models (Hoogeboom et al., 2021) and D3PMs (Austin et al., 2021). We carefully design a forward noising process that allows for tractable sampling of corrupted sequences and computing estimates of the log-likelihood bound. We show that our models outperform in-place diffusion for modeling arithmetic sequences, and that for text they learn error-correction mechanisms that work on misaligned inputs.

## 2. Background

Here we describe previous work that is needed to introduce our method; see Appendix A for additional related work.

### 2.1. Denoising diffusion probabilistic models

DDPMs are latent variable generative models defined by a forward Markov process $q(\boldsymbol{x}_t|\boldsymbol{x}_{t-1})$ which gradually adds

noise, and a learned reverse process $p_\theta(x_{t-1}|x_t)$ that removes noise. The forward process defines a joint distribution $q(x_{0:T}) = q(x_0) \prod_{t=1}^T q(x_t|x_{t-1})$ where $q(x_0)$ is the data distribution and $x_1, x_2, ..., x_T$ are increasingly noisy latent variables that converge to a known distribution $q(x_T)$. The reverse process $p_\theta(x_{t-1}|x_t)$ is then trained to match the forward process posteriors $q(x_{t-1}|x_t, x_0)$, yielding a gradual denoising model with a tractable variational bound on the log-likelihood. To enable efficient training, $q$ is often chosen such that these posteriors can be computed analytically. For continuous DDPMs, $q(x_{t-1}|x_t)$ is typically a Gaussian. For discrete DDPMs, Hoogeboom et al. (2021) propose setting $q$ to a mixture of a uniform distribution and a point mass at the previous value, and Austin et al. (2021) consider using a wider class of structured Markov transition matrices. All recent diffusion models perform corruption in-place: the $k$th element of $x_t$ is a noisier version of the $k$th element of $x_{t-1}$, with no dependence on other tokens.

### 2.2. Levenshtein and Insertion-Deletion Transformers

The Levenshtein Transformer (Gu et al., 2019) learns to insert and delete tokens over a series of generation steps. In each step, it marks tokens in the current sequence $x$ that should be deleted, predicts how many tokens should be inserted at each position, and finally predicts values for the newly inserted tokens. It is trained to imitate the optimal sequence of edit actions computed by a dynamic program in order to recover the dataset example $x$ from a noisy proposal $x'$ (generated by corrupting $x$ or sampling from the model).

The Insertion-Deletion Transformer (Ruis et al., 2020) uses a sequence of insertion steps followed by a single deletion phase. In each insertion step, it takes a random subsequence $x'$ of the original sequence $x$, and learns to insert at most one token between each element of $x'$ according to a random generation order of $x$ from $x'$. In the deletion phase, it takes a (possibly perturbed) proposal from the insertion phase and learns to delete any token that is not part of $x$.

Both of these approaches have focused on the sequence-to-sequence setting, where there are usually only a small set of possible correct answers. Additionally, neither provide a tractable estimate of the log-likelihood of dataset samples under the model; they are instead trained using hand-designed losses for insertion and deletion phases.

## 3. Method

Our goal is to design an insertion-deletion-based generative model within the probabilistic framework of diffusion models with a tractable bound on the log-likelihood. The main considerations are (a) how to define the forward corruption process so that it leads to a reverse process with insertions, deletions, and replacements, (b) how to parameterize the

reverse process, and (c) how to do both tractably within the diffusion process framework.

### 3.1. Forward Process

The forward corruption process specifies how to gradually convert data $x_0$ into noise $x_T$ by repeatedly applying a single-step forward process $q(x_t|x_{t-1})$. Since the learned reverse process is trained to undo each of these corruption steps, and insertion and deletion are inverses, we can obtain a learned reverse process with deletion, insertion, and replacement operations by including insertion, deletion, and replacement operations in the forward process, respectively.

A challenge is that if a single forward step can apply an arbitrary set of insertions, deletions, and replacements, then there may be many ways to get $x_t$ from $x_{t-1}$. For example, $x_t$ can be related to $x_{t-1}$ through the minimum edit between the two, or by deleting the full $x_{t-1}$ and then inserting the full $x_t$. In order to compute $q(x_t|x_{t-1})$, one would need to sum over all these possibilities. To avoid this, we restrict the forward process so that there is a single way to get each $x_t$ from each $x_{t-1}$, by adding two auxiliary symbols into the vocabulary that explicitly track insertion and deletion operations: every insertion operation produces the insertion-marker token $\boxed{\text{INS}}$, and every deletion operation deletes the deletion-marker token $\boxed{\text{DEL}}$. (We note that, since the reverse process is *reversing* the forward corruption process, the learned model must instead insert $\boxed{\text{DEL}}$ and delete $\boxed{\text{INS}}$.) We propose the following form for $q(x_t|x_{t-1})$:

1. Remove all $\boxed{\text{DEL}}$ tokens from $x_{t-1}$.
2. For each token $x$ in $x_{t-1}$, sample a new value (possibly $\boxed{\text{DEL}}$) as $x' \sim \text{Cat}(x'; \delta_x^T Q_t)$, where $Q_t$ is a Markov transition matrix and $\delta_x$ is a one-hot vector for $x$.
3. Between each pair of tokens in the result, and also at the start and end of the sequence, sample $n_i^{\text{new}} \sim \text{Geom}(1 - \alpha_t)$ and insert that many $\boxed{\text{INS}}$ tokens. (We explain this choice in Section 3.4.)

We allow $Q_t$ to include transitions from $\boxed{\text{INS}}$ to any other token, and from any token to $\boxed{\text{DEL}}$, but disallow transitions to $\boxed{\text{INS}}$ or from $\boxed{\text{DEL}}$ to ensure they only arise from insertions and deletions. This ensures unique 1-step alignments.

### 3.2. Parameterization of the reverse process

As an inductive bias, we prefer reverse processes that produce $x_{t-1}$ by modifying $x_t$, instead of predicting it from scratch. As such, the learned reverse process $p_\theta(x_{t-1}|x_t)$ first removes all $\boxed{\text{INS}}$ tokens from $x_t$, then predicts two things for each remaining token: the previous value of the token (which might be $\boxed{\text{INS}}$ if the token should be removed), and the number of $\boxed{\text{DEL}}$ tokens that should be inserted before the token. (Recall that, since this is the *reverse* process, the auxiliary tokens have opposite meanings here.) We also take

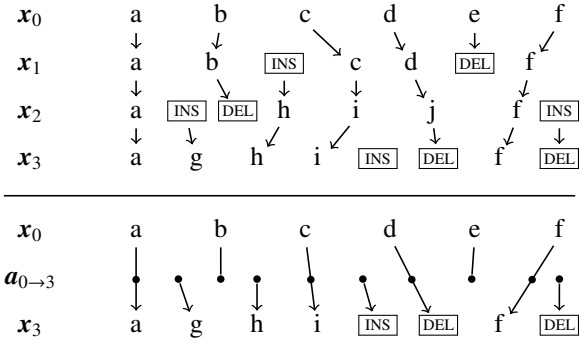

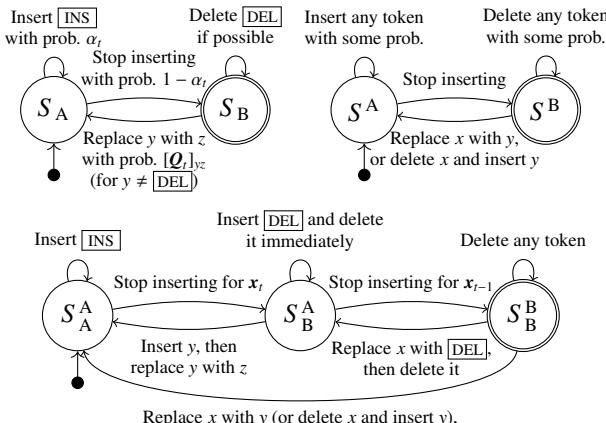

*Figure 2.* An example of sequences $x_0$ through $x_3$ produced by a forward process $q(x_t|x_{t-1})$ (top), along with the corresponding edit summary $a_{0\to3}$ (bottom) that summarizes how to obtain $x_t$ from $x_0$ without describing the full sample path. Note that multiple sample paths can correspond to the same edit summary. Our model $p_\theta$ predicts the corresponding $\boxed{v \text{—}\bullet\text{—}}$ or $\boxed{\bullet\text{—}}$ edge in $a_{0\to t}$ for each token in $x_t$ (including the previous value $v$ in the first case), and also predicts the number of $\boxed{\text{—}\bullet}$ edges immediately before each token in $x_t$ (e.g. there is one before 'f' and zero before 'i').

inspiration from other work on diffusion models (Ho et al., 2020; Hoogeboom et al., 2021), which find improved performance by guessing $x_0$ and then using knowledge of the forward process to derive $p_\theta(x_{t-1}|x_t)$, as opposed to specifying $p_\theta(x_{t-1}|x_t)$ directly. Our full parameterization combines these two ideas: it attempts to infer the edit summary $a_{0\to t}$ that was applied to $x_0$ to produce $x_t$ (as shown in Fig. 2), then uses the known form of $q(x_{t-1}|x_t, x_0, a_{0\to t})$ to derive $p_\theta(x_{t-1}|x_t)$. Specifically, we compute

$$p_\theta(x_{t-1}|x_t) \propto \sum_{\widetilde{x}_0, \widetilde{a}_{0\to t}} \widetilde{p}_\theta(\widetilde{x}_0, \widetilde{a}_{0\to t}|x_t) \cdot q(x_t, x_{t-1}, \widetilde{a}_{0\to t}|\widetilde{x}_0), \quad (1)$$

where tildes denote predictions that are not directly supervised, and we intentionally use $q(x_t, x_{t-1}, \widetilde{a}_{0\to t}|\widetilde{x}_0)$ in place of $q(x_{t-1}|x_t, \widetilde{x}_0, \widetilde{a}_{0\to t})$ to prevent the model from predicting edits $\widetilde{a}_{0\to t}$ that have zero probability under $q(x_t, a_{0\to t}|x_0)$. Intuitively, the model predicts a summary of which edits likely happened (at an unknown time $s \le t$) to produce $x_t$, then $q$ determines the details of which specific edits appeared in $x_{t-1}$. This parameterization requires us to be able to compute $q(x_t, x_{t-1}, \widetilde{a}_{0\to t}|\widetilde{x}_0)$ (discussed in Section 3.4).

### 3.3. Loss function

We optimize the standard evidence bound on the negative log-likelihood, which can be expressed as

$$L = \mathbb{E}_{q(x_{0:T})}\left[\underbrace{-\log p_\theta(x_T)}_{L_T} + \sum_{t=1}^{T} \underbrace{-\log \frac{p_\theta(x_{t-1}|x_t)}{q(x_t|x_{t-1})}}_{L_{t-1}}\right]. \quad (2)$$

For the $L_{t-1}$ terms, we randomly sample $t$ and then compute

$$\mathbb{E}_{q(x_t, x_0, a_{0\to t})}\left[\mathbb{E}_{q(x_{t-1}|x_t, x_0, a_{0\to t})}\left[-\log \frac{p_\theta(x_{t-1}|x_t)}{q(x_t|x_{t-1})}\right]\right]. \quad (3)$$

*Figure 3.* Representation of $q(x_t|x_{t-1})$ (left) and $q(x_{t-1}|x_0)$ (right) as PFSTs, along with their composition $q(x_t, x_{t-1}|x_0)$ (bottom). Execution starts at the black dot and continues until reaching end-of-sequence at the double-outlined state. Some probabilities omitted for readability; see Fig. 5 (in Appendix B) for details.

It turns out that we can compute this inner expectation in closed form given $(t, x_0, x_t, a_{0\to t})$ (see Section 3.4).

For the $L_T$ term, we choose $q$ so that $q(x_T|x_{T-1})$ deterministically replaces every token with $\boxed{\text{DEL}}$ and inserts no new tokens; this implies $x_T$ will always consist of repetitions of $\boxed{\text{DEL}}$, so we can simply learn a tabular distribution $p_\theta(|x_T|)$ of final forward process lengths.

### 3.4. Computational considerations

While a diffusion model could be trained by simply drawing sequences $x_0, x_1, \ldots, x_T$ and training the model to undo each step, these models are usually trained by analytically computing the $L_{t-1}$ terms for individual timesteps $t$ and samples $(x_0, x_t)$, by using closed form representations of $q(x_t|x_0)$ and $q(x_{t-1}|x_t, x_0)$ (Ho et al., 2020). Unfortunately, doing this for a forward process that inserts and deletes tokens is nontrivial. Over multiple steps, the $\boxed{\text{INS}}$ and $\boxed{\text{DEL}}$ markers may be skipped, which means that (as mentioned in Section 3.1) there will likely be many possible sets of insertions and deletions that produce $x_t$ from $x_0$, with a corresponding wide variety of intermediate sequences $(x_1, x_2, \ldots, x_{t-1})$.

To address this challenge, we introduce two main ideas: (a) cast the necessary quantities in terms of *probabilistic finite-state transducers* (PFSTs), which allow us to marginalize out details about intermediate sequences that do not matter for computing the loss, and (b) choose to condition on the edit summary $a_{0\to t}$ in addition to $(x_0, x_t)$ while analytically computing the loss term $L_{t-1}$ in Eq. (3), which allows us to efficiently compute those PFST-based quantities.

A PFST is a probabilistic finite state machine that has an input tape and one or more output tapes. It repeatedly makes stochastic transitions based on a set of transition probabili-

*Figure 4.* Left: generating text with an insertion-deletion denoising model $p_\theta(x_{t-1}|x_t)$ trained on the text8 dataset (generative process flows upward). Right: Fixing typos using an insert-delete model (and an in-place baseline), showing five random predictions from each model.

|            | NLL (nats)        | Error rate (%)   |
|------------|-------------------|------------------|
| In-place   | $\leq 39.95 \pm 0.06$ | $13.12 \pm 2.40$ |
| 0.4 ins/del| $\leq 36.35 \pm 0.07$ | $5.70 \pm 0.37$  |
| 0.6 ins/del| $\leq 35.71 \pm 0.04$ | $5.16 \pm 0.27$  |
| 0.8 ins/del| $\leq 38.51 \pm 0.17$ | $6.48 \pm 0.13$  |

*Table 1.* Results on arithmetic sequences. NLL denotes negative log-likelihoods, error rate denotes the fraction of the step sizes in each generated example that are different from the most common step size. Standard deviation taken over five random seeds.

ties and the current symbol from the input tape. As it makes transitions, it consumes input tape symbols and writes to its output tape(s). In our case, we begin by expressing $q(x_t|x_{t-1})$ as a PFST, which is possible because geometric random variables can be sampled as a repeated coin flip. This PFST iteratively consumes the input ($x_{t-1}$), transitioning between states and writing to the output ($x_t$). We additionally make use of an algebra over PFSTs that allows composing PFSTs and integrating out output tapes. By composing PFSTs for $q(x_t|x_{t-1})$ and $q(x_{t-1}|x_0)$, we obtain a two-output tape PFST for $q(x_t, x_{t-1}|x_0)$, with which we can integrate out $x_{t-1}$ to obtain $q(x_t|x_0)$. Fig. 3 shows the high-level structure of each PFST; full details are in Appendix B.2.

Given a specific edit summary $a_{0\to t}$, we can reconstruct the state transitions in the PFST for $q(x_t, x_{t-1}|x_0)$, which allows us to compute $q(x_t, x_{t-1}, a_{0\to t}|x_0)$ and $q(x_{t-1}|x_t, x_0, a_{0\to t})$ in closed form. Details on how to compute the necessary terms for our loss in Section 3.3 and our model parameterization in Section 3.2 are given in Appendix B.3 and B.4, respectively.

## 4. Experiment: Toy sequence datasets

We start by exploring the expressive power of our model on a toy dataset of arithmetic sequences. We take a 10-step multinomial diffusion corruption process (Hoogeboom et al., 2021) and augment it with varying probabilities of insertion and deletion. As shown in Table 1, moderate insertion/deletion probabilities lead to better log-likelihoods and to generated sequences with fewer deviations from be-

ing a valid arithmetic sequence. However, if insertions and deletions are too frequent, the noise overpowers the patterns in the data, leading to lower accuracy. Figure 1 shows a sequence generated by the 0.6 insert/delete rate model. See Appendix C.2 for experiment details.

## 5. Experiment: Text generation

We also investigate training a 32-step multinomial-diffusion-based model augmented with insertion and deletion on the character-level language dataset text8 (Mahoney, 2011). Although insert/delete models have slightly worse log-likelihood bounds on this dataset (see Table 2 in App. C), the samples are still high quality, and the models show qualitative differences in the generative process: they can correct spelling errors, insert spaces between words, and make other human-like edits. In Fig. 4 we show a generated sentence from an insert-delete model, and also show that this model can be used to "spellcheck" a badly-human-written sentence without being trained on this task by simply treating the sentence as $x_{10}$ and sampling from $p_\theta(x_0|x_{10})$. The insert-delete model generates imperfect but intuitive suggestions whereas an in-place model generates nonsense due to misalignment issues. See Appendix C.3 for experiment details.

## 6. Discussion

In this work we have opened up the class of denoising-based generative models to more flexible processes that include insertion and deletion in addition to in-place replacements. While we have motivated these models from the perspective of text generation, this class of models could be useful for several other applications, such as image super-resolution (by inserting and deleting pixel rows and columns), video generation (by inserting and deleting frames), and molecular structure generation (by editing SMILES representations (Weininger, 1988)). We are also excited about the potential for incorporating other types of non-in-place edits (such as duplication or reordering) into corruption processes as a strategy for improving denoising-based generative models.

## Acknowledgements

We would like to thank Hugo Larochelle, David Bieber, and Disha Shrivastava for helpful discussions and feedback, and William Chan and Mohammad Norouzi for useful context regarding non-autoregressive sequence models. We would also like to thank Tim Salimans and the anonymous INNF reviewers for reading earlier drafts of this manuscript and giving suggestions for improvement.

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

## A. Other related work

A few other works have studied diffusion-like generative models for structured data, including Seff et al. (2019), which exploits a structured forward process $q(x_t|x_{t-1})$ to impose constraints on the generated samples, and Chan et al. (2020), which iteratively refines an output sequence jointly with its alignment to an input sequence. A number of other edit-based generative models have been proposed, including Guu et al. (2018) which edits prototypical examples in a latent space. In natural language processing, edit-based models have been proposed for learning to simplify complex sentences into simple ones (Alva-Manchego et al., 2017; Dong et al., 2019). In source code applications, it is common to generate edits for bug-fixing (Yin et al., 2019; Zhao et al., 2019; Dinella et al., 2020; Yao et al., 2021). There are also models that use edit distances for purposes of supervision (either directly or via imitation learning), but still generate left-to-right (Graves et al., 2006; Bahdanau et al., 2015; Sabour et al., 2018).

## B. Computing probabilities with PFSTs

In this section we describe how to compute the necessary probabilities for the forward process and learned reverse process using probabilistic finite state transducers.

### B.1. Notation for PFST representations

We will begin by introducing additional notation which will be useful for representing PFSTs of the multi-step forward process probabilities $q(x_{t-1}|x_0)$ and $q(x_t, x_{t-1}|x_0)$.

For all of our PFSTs, we associate each transition with a label "$p : x \mapsto y$", which indicates that, conditioned on $x$ being the next symbol on the input tape, with probability $p$ the PFST consumes $x$ and produces $y$. We use $\varepsilon$ to denote the empty sequence, and thus $p : \varepsilon \mapsto y$ denotes a transition that (with probability $p$) inserts $y$ without consuming any input. Similarly $p : x \mapsto \varepsilon$ denotes consuming $x$ without producing any output, which corresponds to a deletion. For the product transducer $q(x_t, x_{t-1}|x_0)$, we write $p : x \mapsto y \mapsto z$ to indicate consuming $x$ from $x_0$, writing $y$ to $x_{t-1}$, and writing $z$ to $x_t$.

As stated in Section 3.1, each single step of the forward process is parameterized by a scalar $\alpha_t$ and a Markov transition matrix $Q_t$. To represent the aggregate probabilities over multiple steps, we introduce three parameters $\overline{\alpha}_t$, $\overline{\beta}_t$, and $\overline{Q}_t$:

- $\overline{\alpha}_t$ is a vector of insertion probabilities, such that $[\overline{\alpha}_t]_i$ gives the chance of inserting token $i$ when skipping from time 0 to time $t$. In particular, $[\overline{\alpha}_t]_{\langle\text{INS}\rangle}$ denotes the probability of inserting $\boxed{\text{INS}}$, and $[\overline{\alpha}_t]_{\langle\text{DEL}\rangle}$ denotes the probability of inserting $\boxed{\text{DEL}}$. $\overline{\alpha}_t$ is used to summarize inserts at some time $s \leq t$ followed by a chain of replacements $Q_{s+1}, \ldots, Q_t$. If $s < t$, we call this a

*silent insertion.*
- Conversely, $\overline{\beta}_t$ is a vector of deletion probabilities, such that $[\overline{\beta}_t]_i$ gives the chance of deleting token $i$ conditional on it appearing in $x_0$. $\overline{\beta}_t$ is used to summarize a chain of replacements $Q_1, \ldots, Q_s$ that produce $\boxed{\text{DEL}}$ at some time $s < t$. We call this a *silent deletion.*
- Finally, $\overline{Q}_t$ is a matrix that specifies how tokens will be replaced over multiple steps, such that $[\overline{Q}_t]_{xy}$ denotes the probability of consuming $x$ and producing $y$ conditioned on $x$ appearing in $x_0$. Notably, this encompasses both chains of replacements due to $Q_t$, as well as *silent deletion-insertion pairs*, where a token is inserted immediately after a deleted token. (For instance, in Fig. 2, 'b' and 'h' form a deletion-insertion pair)

Using this, we can fully specify the PFSTs for each process of interest:

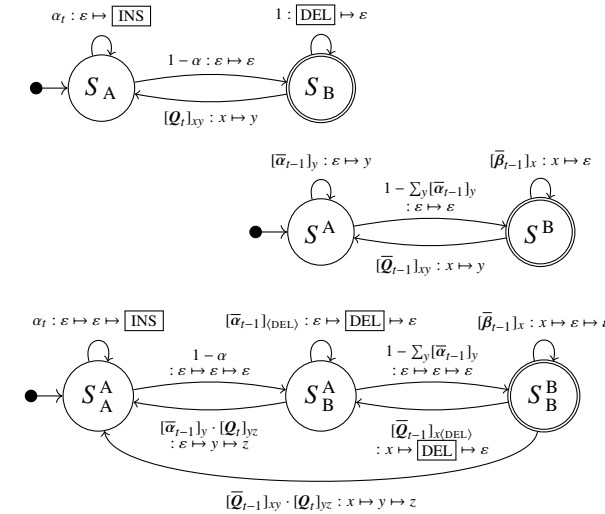

*Figure 5.* From top to bottom: $q(x_t|x_{t-1})$, $q(x_{t-1}|x_0)$, and $q(x_t, x_{t-1}|x_0)$ as probabilistic finite-state transducers.

### B.2. Calculating $q(x_t|x_0)$ from $q(x_t|x_{t-1})$

As discussed in Section 3.4, we can use the transducer representations shown in Fig. 5 to recursively construct probabilities for $q(x_t|x_0)$ from the individual step distributions $q(x_t|x_{t-1})$. We proceed inductively by constructing a deterministic $q(x_0|x_0)$ and then repeatedly computing $q(x_t|x_0)$ from $q(x_{t-1}|x_0)$ and $q(x_t|x_{t-1})$.

As our base case, observe that $q(x_0|x_0)$ is the identity transformation, and we can represent it using the following parameters:

$$[\overline{\alpha}_0]_i = 0, \qquad [\overline{\beta}_0]_i = 0,$$
$$[\overline{Q}_0]_{ij} = 1 \text{ if } i = j, 0 \text{ otherwise.} \tag{4}$$

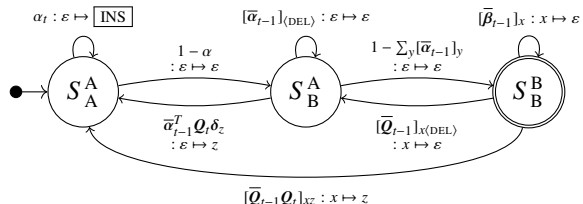

*Figure 6.* Transducer for $q(x_t|x_0)$ after marginalizing out $x_{t-1}$ from $q(x_t, x_{t-1}|x_0)$ in Fig. 5. Note the presence of matrix-vector products with $\overline{\alpha}_{t-1}$ and $\overline{Q}_{t-1}$, instead of explicit indices.

Now suppose we know $\overline{\alpha}_{t-1}$, $\overline{\beta}_{t-1}$, and $\overline{Q}_{t-1}$ for $q(x_{t-1}|x_0)$, and $Q_t$ and $\alpha_t$ for $q(x_t|x_{t-1})$, and we wish to compute $\overline{\alpha}_t$, $\overline{\beta}_t$, and $\overline{Q}_t$ for $q(x_t|x_0)$. We start by constructing the product transducer for $q(x_t, x_{t-1}|x_0)$ by composing the two transducers for $q(x_{t-1}|x_0)$ and $q(x_t|x_{t-1})$, as shown in Fig. 5. Next, we marginalize out the middle timestep $x_{t-1}$. This entails removing the middle step from each transition, and instead summing over all possible values for that middle token. We obtain the two-tape transducer shown in Fig. 6.

Next, we eliminate the middle state $S_B^A$, by replacing all paths that pass through it with new transitions that directly connect $S_A^A$ and $S_B^B$. We note that these paths may enter the loop $S_B^A \mapsto S_B^A$ arbitrarily many times without producing any output or consuming any input (this is a *silent-insertion-deletion pair*). The total probability of all paths that take that loop an arbitrary number of times is thus

$$\sum_{n=0}^{\infty} \left([\overline{\alpha}_{t-1}]_{\langle\text{DEL}\rangle}\right)^n = \frac{1}{1 - [\overline{\alpha}_{t-1}]_{\langle\text{DEL}\rangle}}. \qquad (5)$$

We obtain the following new transitions. From $S_A^A$ to $S_B^A$:

$$(1 - \alpha_t)\frac{1}{1-[\overline{\alpha}_{t-1}]_{\langle\text{DEL}\rangle}}\overline{\alpha}_{t-1}^T Q_t \delta_y : \varepsilon \mapsto y \qquad (6)$$

From $S_A^A$ to $S_B^B$:

$$(1 - \alpha_t)\frac{1}{1-[\overline{\alpha}_{t-1}]_{\langle\text{DEL}\rangle}}(1 - \sum_y[\overline{\alpha}_{t-1}]_y) : \varepsilon \mapsto \varepsilon \qquad (7)$$

From $S_B^B$ to $S_B^B$:

$$[\overline{Q}_{t-1}]_{x\langle\text{DEL}\rangle}\frac{1}{1-[\overline{\alpha}_{t-1}]_{\langle\text{DEL}\rangle}}(1 - \sum_y[\overline{\alpha}_{t-1}]_y : x \mapsto \varepsilon \qquad (8)$$

From $S_B^B$ to $S_A^A$:

$$[\overline{Q}_{t-1}]_{x\langle\text{DEL}\rangle}\frac{1}{1-[\overline{\alpha}_{t-1}]_{\langle\text{DEL}\rangle}}\overline{\alpha}_{t-1}^T Q_t \delta_z : x \mapsto z \qquad (9)$$

Equation (9) is particularly notable, as it corresponds to a *silent-deletion-insertion pair*, in which $q(x_{t-1}|x_0)$ replaces $x$ with DEL and then inserts some other token ($y$ in Fig. 5, but marginalized out here), after which $q(x_t|x_{t-1})$ removes DEL and produces $z$ from $y$.

Combining these new transitions with the old ones between $S_A^A$ and $S_B^B$ gives us the following values for $q(x_t|x_0)$:

$$\overline{\alpha}_t^T = \alpha_t\delta_{\langle\text{INS}\rangle}^T + \frac{1 - \alpha_t}{1 - [\overline{\alpha}_{t-1}]_{\langle\text{DEL}\rangle}}\overline{\alpha}_{t-1}^T Q_t, \qquad (10)$$

$$\overline{\beta}_t = \overline{\beta}_{t-1} + \overline{Q}_{t-1}\delta_{\langle\text{DEL}\rangle}\frac{1 - \sum_y[\overline{\alpha}_{t-1}]_y}{1 - [\overline{\alpha}_{t-1}]_{\langle\text{DEL}\rangle}}, \qquad (11)$$

$$\overline{Q}_t = \overline{Q}_{t-1}Q_t + \frac{\overline{Q}_{t-1}\delta_{\langle\text{DEL}\rangle}\overline{\alpha}_{t-1}^T Q_t}{1 - [\overline{\alpha}_{t-1}]_{\langle\text{DEL}\rangle}} \qquad (12)$$

(Note: Here we assume $[Q_t]_{\langle\text{DEL}\rangle\,i} = [Q_t]_{i\,\langle\text{INS}\rangle} = 0$, as $Q_t$ does not allow transitions from DEL or to INS.) Intuitively, Eq. (10) says that inserts occur either as INS-marked inserts at time $t$ or (silent) inserts before time $t$ that are then perturbed; Eq. (11) says that deletions occur either as silent deletions before time $t$ or as transitions to DEL at time $t$ that are then removed without inserting new tokens; and Eq. (12) says that replacements occur either because a token was copied/replaced before time $t$ and then copied/replaced again at $t$, or because a token $x$ was replaced by DEL at time $t - 1$, but a new token $y$ was (silently) inserted at or before time $t - 1$, so that at time $t$ the new token $y$ looks like a replacement for the old token $x$.

### B.3. Closed form of $q(x_{t-1}|x_t, x_0, a_{0\to t})$

We can similarly obtain a closed-form representation of $q(x_{t-1}|x_t, x_0, a_{0\to t})$ by reasoning backwards about the elimination steps in the previous section. We start by observing that the edit summary $a_{0\to t}$ tells us the sequence of replacements $x \mapsto z$, insertions $\varepsilon \mapsto z$, and deletions $x \mapsto \varepsilon$ executed by the transducer while sampling $x_t$ from $x_0$.

Suppose we observe a replacement $x \mapsto z$ (where perhaps $x = z$ if it was copied unmodified). This must have been produced by the $\overline{Q}_t$ edge. From Eq. (12) and Fig. 5 we can infer the distribution over the intermediate value $x \mapsto y \mapsto z$, if it exists:

$$p(x \mapsto y \mapsto z|x \mapsto z) = \frac{[\overline{Q}_{t-1}]_{xy} \cdot [Q_t]_{yz}}{[Q_t]_{xz}} \qquad (13)$$

$$p\left(\begin{matrix}x \mapsto \boxed{\text{DEL}} \mapsto \varepsilon \\ \varepsilon \mapsto y \mapsto z\end{matrix}\middle|x \mapsto z\right) = \frac{[\overline{Q}_{t-1}]_{x\langle\text{DEL}\rangle}[\overline{\alpha}_{t-1}]_y[Q_t]_{yz}}{(1-[\overline{\alpha}_{t-1}]_{\langle\text{DEL}\rangle})[\overline{Q}_t]_{xz}} \qquad (14)$$

If the event in Eq. (14) occurs, we can also infer that there was a geometric number $n_i^{\text{extra}} \sim \text{Geom}(1 - [\overline{\alpha}_{t-1}]_{\langle\text{DEL}\rangle})$ of extra $\varepsilon \mapsto \boxed{\text{DEL}} \mapsto \varepsilon$ transitions due to the loop in $S_B^A$.

Now suppose we observe an insert $\varepsilon \mapsto z$. If $z = \boxed{\text{INS}}$, we know it was inserted at time $t$, so it must have been produced by the $\varepsilon \mapsto \varepsilon \mapsto \boxed{\text{INS}}$ transition. If $z$ is any other token, it must have already existed at time $t - 1$, with

$$p(\varepsilon \mapsto y \mapsto z|\varepsilon \mapsto z) = \frac{[\overline{\alpha}_{t-1}]_y \cdot [Q_t]_{yz}}{[\overline{\alpha}_{t-1}Q_t]_z}. \qquad (15)$$

In this second case we also pass through $S_B^A$ and generate $n_i^{\text{extra}} \sim \text{Geom}(1 - [\overline{\boldsymbol{\alpha}}_{t-1}]_{\langle\text{DEL}\rangle})$ extra $\varepsilon \mapsto \boxed{\text{DEL}} \mapsto \varepsilon$ transitions.

Next suppose we observe a deletion $x \mapsto \varepsilon$ (where we know $x \neq \boxed{\text{DEL}}$ because there are no deletion markers in the data distribution). In this case we have

$$p(x \mapsto \varepsilon \mapsto \varepsilon | x \mapsto \varepsilon) = \frac{[\overline{\boldsymbol{\beta}}_{t-1}]_x}{[\overline{\boldsymbol{\beta}}_t]_x} \quad (16)$$

$$p(x \mapsto \boxed{\text{DEL}} \mapsto \varepsilon | x \mapsto \varepsilon) = \frac{[\overline{\boldsymbol{Q}}_{t-1}]_{x\langle\text{DEL}\rangle}^{\frac{1 - \sum_y [\overline{\boldsymbol{\alpha}}_{t-1}]_y}{1 - [\overline{\boldsymbol{\alpha}}_{t-1}]_{\langle\text{DEL}\rangle}}}}{[\overline{\boldsymbol{\beta}}_t]_x} \quad (17)$$

where, like before, the second case passes through $S_B^A$ and generates $n_i^{\text{extra}} \sim \text{Geom}(1 - [\overline{\boldsymbol{\alpha}}_{t-1}]_{\langle\text{DEL}\rangle})$ extra $\varepsilon \mapsto \boxed{\text{DEL}} \mapsto \varepsilon$ transitions.

Finally, we note that every time we move from $S_A^A$ to $S_B^B$ (in other words, whenever we stop inserting tokens), there is one more $n_i^{\text{extra}} \sim \text{Geom}(1 - [\overline{\boldsymbol{\alpha}}_{t-1}]_{\langle\text{DEL}\rangle})$ set of $\varepsilon \mapsto \boxed{\text{DEL}} \mapsto \varepsilon$ transitions.

Using the above analysis allows us to compute $q(\boldsymbol{x}_{t-1}, \boldsymbol{a}_{0\to(t-1)} | \boldsymbol{x}_t, \boldsymbol{x}_0, \boldsymbol{a}_{0\to t})$, where the extra information $\boldsymbol{a}_{0\to(t-1)}$ specifies the sequence of $x \mapsto \varepsilon \mapsto \varepsilon$, $\varepsilon \mapsto \boxed{\text{DEL}} \mapsto \varepsilon$ and $x \mapsto \boxed{\text{DEL}} \mapsto \varepsilon$ transitions (which are ambiguous from $\boldsymbol{a}_{0\to t}$ alone). Since we do not particularly care about this information, we can marginalize it out by noting that the total number $n^{\text{obs}}$ of consecutive $\boxed{\text{DEL}}$ tokens observed at a particular position in $\boldsymbol{x}_{t-1}$ is the sum of the number of explicit deletions $x \mapsto \boxed{\text{DEL}} \mapsto \varepsilon$ and insertion-deletion pairs $\varepsilon \mapsto \boxed{\text{DEL}} \mapsto \varepsilon$. Given a fixed number of explicit deletions, the total number of insertion-deletion pairs is a sum of independent geometric random variables and thus has a negative binomial distribution. We can thus:

- compute for each deleted token $x$ in $\boldsymbol{x}_0$ the probability of an explicit $x \mapsto \boxed{\text{DEL}} \mapsto \varepsilon$ transition using Eq. (17)
- compute for each perturbed $x \mapsto z$ transition the probability of an explicit $x \mapsto \boxed{\text{DEL}} \mapsto \varepsilon$ transition using Eq. (14)
- compute the distribution of the total number $n^{\text{explicit}}$ of $x \mapsto \boxed{\text{DEL}} \mapsto \varepsilon$ transitions at this location in $\boldsymbol{x}_{t-1}$ by noting that it is a sum of independent Bernoulli r.v.s (which can be computed either by taking convolutions of their PMFs, or, if all tokens are deleted with the same probability, by observing that this is a binomial distribution)
- use this distribution to compute a mixture of negative binomial distributions: $n^{\text{obs}} \sim n^{\text{explicit}} + \text{NB}(n^{\text{explicit}} + 1, 1 - [\overline{\boldsymbol{\alpha}}_{t-1}]_{\langle\text{DEL}\rangle})$.

## B.4. Combining $\widetilde{p}_\theta(\widetilde{\boldsymbol{x}}_0, \widetilde{\boldsymbol{a}}_{0\to t} | \boldsymbol{x}_t)$ with $q$

The $\boldsymbol{x}_0$-predicting parameterization of $p_\theta$ follows the same general procedure outlined above for inferring $\boldsymbol{x}_{t-1}$ from

$\boldsymbol{x}_t, \boldsymbol{x}_0$ and $\boldsymbol{a}_{0\to t}$. However, we make a few slight modifications due to the structure of $\widetilde{p}_\theta$.

For each token $z$ in $\boldsymbol{x}_t$, the model predicts a modification probability $\widetilde{p}_\theta(x \mapsto z)$ for each token and an insertion probability $\widetilde{p}_\theta(\epsilon \mapsto z)$. We use these as weights to scale the appropriate inference terms in Eqs. (13) to (15).

Additionally, the model predicts a distribution $\widetilde{p}_\theta(n_i^{del})$ of the number $x \mapsto \varepsilon$ transitions that occurred before each position $i$ in $\boldsymbol{x}_t$. We use this to infer the number $n_i^{\text{obs}}$ of $\boxed{\text{DEL}}$ placeholders that appear at time $t-1$ using the same inference procedure as above, but we now have a *mixture* of mixtures of negative binomial distributions because we may be uncertain about how many insertions there were. (Usually, we will have $n_i^{\text{obs}} \leq n_i^{del}$, since deletions could have occurred at any time from 0 to $t$.) When implementing this parameterization we assume that every token is equally likely to be deleted at each timestep, so that the model only has to predict the number of missing tokens from $\boldsymbol{x}_0$; if this is not the case, it would be possible to predict $\widetilde{p}_\theta(n_i^{obs})$ directly instead.

We choose to predict deletion-insertion pairs simply as an insertion preceded by a deletion, instead of reasoning about it as a replacement; this simplifies our computation by avoiding having to separately reason about Eq. (14).

## C. Experimental details

### C.1. Model architecture

For all of our experiments, we use a standard decoder-only transformer following the T5 (Raffel et al., 2020) architecture, with either six or twelve layers depending on the task. The main modification we make is to introduce two output heads instead of one. The first output head, like a standard transformer, predicts a matrix $f_\theta(\boldsymbol{x}_t) \in \mathbb{R}^{L \times K}$ of unnormalized log-probabilities (logits), where $L$ is the sequence length and $K$ is the vocabulary size. We interpret $f_\theta(\boldsymbol{x}_t)_{iv}$ as the log-probability of the $i$th token being produced by a replacement edit $\boxed{v \to \bullet}$ (equivalently $v \mapsto [\boldsymbol{x}_t]_i$ in the PFST notation) in the edit summary $\boldsymbol{a}_{0\to t}$, and similarly interpret $f_\theta(\boldsymbol{x}_t)_{i\langle\text{INS}\rangle}$ as the log-probability of the $i$th token of $\boldsymbol{x}_t$ being an insertion $\boxed{\to\bullet}$ (or $\varepsilon \mapsto [\boldsymbol{x}_t]_i$). We reuse the embeddings for the input vocabulary as the final output layer for this head. The secound output head produces a matrix $g_\theta(\boldsymbol{x}_t) \in \mathbb{R}^{L \times L}$, for which $f_\theta(\boldsymbol{x}_t)_{in}$ gives the (unnormalized) log-probability of having $n$ different $\boxed{\to\bullet}$ (or $[\boldsymbol{x}_0]_j \mapsto \varepsilon$) edges immediately before the $i$th token of $\boldsymbol{x}_t$.

When running the transformer on an input sequence, we introduce an extra end-of-sequence token EOS that denotes the last position in the input. The first output head $f_\theta$ is ignored for the EOS token, but we do use the output $g_\theta$ for the EOS token to determine the number of $\boxed{\to\bullet}$ edges in the

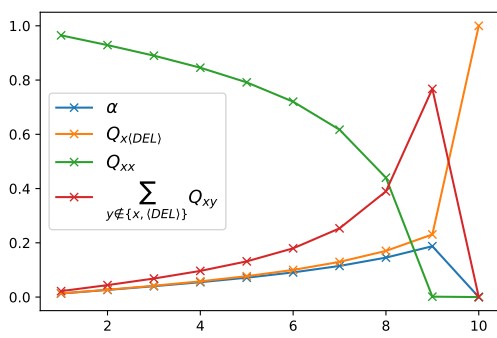

*Figure 7.* Noise schedule for arithmetic sequence task for $r = 0.6$. For each number $x \in \{0, \ldots, 511\}$, probability mass shown by the red line is evenly divided among all of the other 511 dataset tokens (not including $\boxed{\text{INS}}$ or $\boxed{\text{DEL}}$). Schedules for other values of $r$ are similar, but with higher or lower values of $\alpha$ and $\boldsymbol{Q}_{x\langle DEL \rangle}$.

edit summary $\boldsymbol{a}_{0 \to t}$ that occur at the end of the sequence.

As mentioned in Section 3.3, we additionally store a fixed-size table $p_\theta(|\boldsymbol{x}_T|) \in \mathbb{R}^L$, which we fit to the distribution of observed lengths $\boldsymbol{x}_T$.

## C.2. Arithmetic sequences

We construct a dataset of arithmetic sequences by randomly sampling a step size $s$ between 1 and 10, a direction (increasing or decreasing), a length $\ell$ between 32 and 64 (with the constraint that $s(\ell - 1) < 509$), and finally a random starting position so that all terms in the sequence are between 2 and 511, inclusive. (0 is used to denote padding in the data loader, and 1 was reserved for preliminary experiments that required additional reserved tokens, but both are treated as ordinary tokens by the model.) Along with $\boxed{\text{INS}}$, $\boxed{\text{DEL}}$, and an end-of-sequence marker EOS, this yields a total augmented vocabulary of size 515.

We compare four different forward process schedules, each of which is tuned to add less noise for timesteps closer to 0 and more noise as $t$ approaches 10. We start by choosing an insert/delete rate $r \in \{0, 0.4, 0.6, 0.8\}$. Next, for $1 \leq t \leq 9$, we calculate a fraction $u_t = 0.1\frac{t}{9} + 0.9\left(\frac{t}{9}\right)^2$, then choose the insertion probability $\alpha_t$ and matrix $\boldsymbol{Q}_t$ for each $t$ so that, cumulatively after step $t$, approximately $u_t \times r$ of the elements of $\boldsymbol{x}_0$ have been deleted, $u_t \times r$ of the elements of $\boldsymbol{x}_t$ come from insertions (so that the length of the sequence remains approximately the same), and $u_t$ of the remaining elements from $\boldsymbol{x}_0$ have been replaced by a random integer between 0 and 512. Finally, at step 10 we append a deterministic step $\boldsymbol{Q}_{10}$ that replaces every token with $\boxed{\text{DEL}}$, and set $\alpha_{10} = 0$. When $r = 0.0$, no insertions or deletions occur until the last step, which is simply used to allow the model to predict the length of the sequence. We choose $[\boldsymbol{Q}_t]_{\langle INS \rangle n} = \frac{1}{512}$ for all $0 \leq n < 512$ so that $\boxed{\text{INS}}$ is equally likely to transition to any

| | Bits/char |
|---|---|
| In place | $\leq 1.669$ |
| 0.4 insert/delete | $\leq 1.759$ |
| 0.6 insert/delete | $\leq 1.789$ |
| 0.8 insert/delete | $\leq 1.844$ |

*Table 2.* Preliminary quantitative results on text8. Shown are the best results over a hyperparameter sweep of 12 learning rate schedules.

of the 512 numbers in the vocabulary. The full schedule for $r = 0.6$ is shown in Fig. 7.

For each insert/delete rate $r$, we train a six-layer transformer model over 100,000 minibatches of 512 random examples, using the Adam optimizer and a learning rate that increases linearly to $2 \times 10^{-4}$ over 5000 steps, then stays constant. We rerun training with five random seeds for each schedule. Since the loss seemed to stabilize at around 90,000 steps, we take averages of the validation metrics computed during the last 10,000 steps of training for each seed, corresponding to ELBO estimates for 46,080 random dataset examples and error rate metrics for 2304 samples drawn from the model. We then report the average and standard deviation of these per-seed metrics across the five random seeds for each schedule.

## C.3. Text generation on text8

For text8, we construct a dataset of training examples by taking randomly-selected 118 character chunks of the full concatenated lower-cased training set. We use a dataset vocabulary of 28 tokens, including each character 'a' through 'z', a space, and an extra token '-' that does not appear in the dataset (again used for preliminary experiments); including $\boxed{\text{INS}}$, $\boxed{\text{DEL}}$, and EOS gives a vocabulary of size 31. During training, since we may insert a large number of tokens by chance, we enforce a maximum length of the intermediates $\boldsymbol{x}_t$ by rejection sampling until we draw a sample shorter than 128 characters (which we correct for when computing the ELBO during evaluation).

As in the arithmetic sequence dataset, we compare forward process schedules with four insert/delete rates $r \in \{0, 0.4, 0.6, 0.8\}$, constructed to add less noise near time 0. In this case, we instead set $u_t = 0.1\frac{t}{31} + 0.9\left(\frac{t}{31}\right)^2$ and produce a 32-step corruption process; similarly, when randomizing, we randomly choose from the 28 tokens in the vocabulary instead of the 512 numbers.

For each insert/delete rate $r$, we train a twelve-layer transformer model over 1,000,000 minibatches of 512 random examples, using the Adam optimizer. We perform a sweep over four learning rates $\{5 \times 10^{-5}, 1 \times 10^{-4}, 2 \times 10^{-4}, 5 \times 10^{-4}\}$ and three schedule types: linear increase until 5000 steps

followed by constant, linear increase until 5000 steps followed by reciprocal square root decay, and a cyclical cosine schedule with period 100,000.

As a preliminary estimate of performance, and because training seemed to converge before 900,000 steps, we evaluated over a subset of 40,960 length-118 segments sampled from the validation set, averaged over the last 100,000 steps of training. Table 2 shows preliminary bits/char measurements for the run with the best performance for each value of $r$.

To produce the typo-repair example on the right side of Fig. 4, we took the human-written sentence "thisn sentsnetne wasstype vssry babdly", intended as a typo-ridden version of "this sentence was typed very badly". We then padded the sentence out with placeholder text ("lorem ipsum dolor sit amet lorem ipsum dolor sit amet...") until it had length 119, to be approximately the length of the training examples. We set this padded sentence as $x_{10}$, then drew five random samples for both the 0.6 insert/delete rate model and the 0.0 insert/delete rate model. We trimmed off the placeholder text (which the model generally left alone) but did not make any other edits.