# OpenReview forum: "Beyond In-Place Corruption: Insertion and Deletion In Denoising Probabilistic Models"
_ICML.cc/2021/Workshop/INNF — INNF+ 2021 poster_

### Official Review · Reviewer_Unua · 2021-06-11

**Rating:** Borderline Accept
**Confidence:** 1

**Summary:**

The paper extends denoising diffusion probabilistic models by incorporating insertion and deletion operations. The authors propose a generative model with a tractable bound on the log-likelihood. The forward process, which gradually converts data to noise in successive steps, is restricted such that there is exactly one specific way how to get from one variable to another. The reverse process is designed to reverse the forward corruption process.

**Justification For Rating:**

The paper presents an extension that brings an existing framework closer to the framework of normalizing flows. I found the paper interesting in that it implements the idea of normalizing flows in quite a different setting than I'm used to. I think it could therefore inspire other developments and trigger interesting discussions. However, I should point out that this is not my field and it is therefore harder for me to assess the novelty and technical contribution. One thing that I would suggest is that the authors make a stronger connection to one of the more classical normalizing-flow approaches.

---

### Official Review · Reviewer_pW4X · 2021-06-11

**Rating:** Accept
**Confidence:** 3

**Summary:**

This paper proposes a generative denoising probabilistic model which allows insertion and deletion operations in the process. The challenges which are involved are discussed and practical solutions to each are proposed. Two short experiments are also given and allow to explore some features of the proposed model.

**Justification For Rating:**

Incorporating deletions and insertions is a desirable feature in a denoising model and the experiments do show a significant improvement when compared to an other recent model. Moreover, this paper is well written and the ideas proposed are reasonable and allow to construct a meaningful model.

This is a good workshop submission which should be of interest in the community. Improvements in the empirical evaluation of the ideas presented here (more extensive comparisons, other tasks,...) could improve it substantially.

---

### Official Review · Reviewer_rzz1 · 2021-06-12

**Rating:** Accept
**Confidence:** 2

**Summary:**

Autoregressive models present practical limitations for sequence, and in particular language, modeling (e.g. they do not allow for parallel generation).
While Denoising Diffusion Probabilistic Models (DDPMs) provide a more efficient alternative, they impose strong constraints on the generative process, namely they only allow for in-place corruptions. Alternative sequence models including insertions and deletions exist (e.g. the Levenshtein transformer), but they are not formulated as generative models and do not allow for log-likelihood evaluation.

This work introduces a new way to incorporate insertions and deletions within a generative model based on DDPMs, with a tractable bound on the log-likelihood.
The proposed forward (corruption) process specifies a unique possible transition between any $\mathbf{x}_{t-1}$ and $\mathbf{x}_t$, by introducing two “INS” and “DEL” tokens marking, respectively, insertions and deletions, and by suitably specifying a Markov transition matrix (e.g. disallowing transitions from “INS” to “DEL”). The reverse process has a parametrisation inspired by other work on diffusion models. The loss function is an evidence bound on the negative log-likelihood, and requires closed form representations of (forward) terms entering the chosen parameterisation of the reverse process. To this end, the relevant quantities are expressed as probabilistic finite state transducers (PFSTs), with suitable choices for the transition probabilities.

This allows tractable sampling and estimates of log-likelihood, while lifting the constraint of in-place corruptions imposed by standard DDPM models.


**Justification For Rating:**

The submission is very well written and motivated, and I enjoyed reading it.
The authors provide a flexible probabilistic framework to handle deletion and insertion processes in sequence modeling.
Each step, challenge and proposed solution is clearly explained—as much as the workshop submission format allows.
The submission seems solid.

Minor notes:

Sec. 3.1: “our first idea” —— what is the second idea?

Additional comments on the non-monotonic behaviour on NLL and Error rate as a function of insertion/deletion probability in Table 1 would be helpful.

For a future conference paper, a stronger experimental validation and more elaboration on the technical details, particularly in sections 3.2 and 3.4, are recommended.

---

### Decision · Program_Chairs · 2021-06-14

Accept (poster)